# Pediatric Coccidioidal Meningitis: A Systematic Review and Proportional Synthesis of Cases Reported in the Fluconazole Era (2000–2025)

**DOI:** 10.3390/jof11100713

**Published:** 2025-10-01

**Authors:** Maria F. De la Cerda-Vargas, Pedro Navarro-Dominguez, Elizabeth Meza-Mata, Melisa A. Muñoz-Hernandez, Fany Karina Segura-Lopez, Marisela Del Rocio Gonzalez-Martinez, Hector A. Delgado-Aguirre, Sergio Valente Flores-Miranda, David de Jesús Mercado-Rubio, Yair O. Adame-Martínez, Geovanni A. Valadez-Altamira, Jose Antonio Candelas-Rangel

**Affiliations:** 1Department of Neurosurgery and Neurotechnology, Universitätsklinik Tübingen, 72076 Tübingen, Germany; 2Department of Neurosurgery, Medical Specialties Hospital No. 71, Instituto Mexicano del Seguro Social, Torreon 27000, Coahuila, Mexico; 3Department of Pathological Anatomy, Medical Specialties Hospital No. 71, Instituto Mexicano del Seguro Social, Torreon 27000, Coahuila, Mexico; 4Department of Health and Research, Medical Specialties Hospital No. 71, Instituto Mexicano del Seguro Social, Torreon 27000, Coahuila, Mexico; 5Department of Microbiology, Faculty of Medicine, Universidad Autónoma de Coahuila, Torreon 27000, Coahuila, Mexico; 6Department of Transplants, Medical Specialties Hospital No. 71, Instituto Mexicano del Seguro Social, Torreon 27000, Coahuila, Mexico; 7Department of Neurosurgery, Regional General Hospital No. 2, Instituto Mexicano del Seguro Social, Ciudad Juárez 32424, Chihuahua, Mexico; 8Department of Neurosurgery, General Hospital of Zone No. 11, Mexican Social Security Institute (IMSS), Piedras Negras 26070, Coahuila, Mexico

**Keywords:** coccidioidal meningitis, children, hydrocephalus, CNS coccidioidomycosis, pediatric fungal infections, neurococcidioidomycosis

## Abstract

Coccidioidal meningitis (CM) is a rare but life-threatening complication of disseminated coccidioidomycosis, occurring in ~16% of cases, particularly among children in endemic regions such as the southwestern US and northern Mexico. Without timely diagnosis and antifungal therapy, pediatric CM is almost universally fatal within the first year. Hydrocephalus develops in up to 50% of cases. In 2000, Galgiani et al. established fluconazole as first-line therapy for CM. Subsequent guidelines refined management but did not specifically address pediatric patients (>1 month–≤19 years). No studies in the fluconazole era have systematically evaluated risk factors for complications in this population. We therefore conducted a systematic review and proportional synthesis of pediatric CM cases, focusing on CNS complications and outcomes. PubMed/MEDLINE, Embase (Ovid), and Web of Science were systematically searched (2000–2025). PROSPERO registration ID (1130290). Inclusion criteria encompassed epidemiological studies, case series, and case reports that described at least one pediatric case of CM or CNS involvement, confirmed by diagnostic methods. Cases in adults, neonates (<1 month), congenital infections, teratogenicity studies, reviews, or incomplete reports were excluded. Only cases with complete individual data (*n* = 48) were included. Methodological rigor was ensured using JBI Critical Appraisal Tools. Of 1089 studies, 31 met the inclusion criteria, representing 3874 pediatric cases. CM/CNS involvement was confirmed in 165 cases (4.25%; 95% CI: 3.6–4.9%), with hydrocephalus in 62 (37.5%). Among 48 case reports with complete data, fluconazole was first-line therapy in 65%. Serum CF titers ≥ 1:16 were associated with hydrocephalus plus stroke (*p* = 0.027) and independently predicted adverse outcomes (relapse/death; OR = 4.5, *p* = 0.037), whereas lifelong azole therapy was associated with improved outcomes (overall survival mean, 82 vs. 32 months; *p* = 0.002). Pediatric CM remains highly lethal, with hydrocephalus a frequent and severe complication. High serum CF titers (≥1:16) predict poor outcomes, emphasizing the urgent need for standardized, pediatric-specific diagnosis and management guidelines.

## 1. Introduction

Coccidioidal meningitis (CM) is a rare but life-threatening manifestation of coccidioidomycosis, a fungal infection caused by *Coccidioides* spp., endemic to the Southwestern United States, Northern Mexico, and parts of Central and South America [1,2]. In recent years, pediatric CM cases have also been reported outside endemic regions, including Egypt [3] and China [4]. CM represents the most severe form of extrapulmonary dissemination, occurring in up to 16% of such cases [5]. Although pediatric coccidioidomycosis accounts for approximately 5–10% of all reported infections, meningitis in this population remains uncommon, estimated at only 1% of infected children [6,7]. Without timely diagnosis and appropriate therapy, pediatric CM is almost universally fatal within the first year [8].

Infection typically begins with the inhalation of arthroconidia. While most infections are asymptomatic or limited to mild pulmonary disease, a small proportion progress to disseminated disease involving the central nervous system (CNS) [9]. Factors associated with this progression include immunosuppression (e.g., HIV infection, organ transplantation, or immunosuppressive therapy), genetic susceptibility affecting innate immune responses, and delayed or inadequate antifungal treatment [10,11,12]. Coccidioidal meningitis (CM) is particularly severe, often requiring prompt and sometimes lifelong antifungal therapy. Pediatric patients pose unique diagnostic and therapeutic challenges, and delayed recognition due to initial misdiagnosis is common [13].

Management of pediatric CM generally involves high-dose azoles or intrathecal polyenes, with monitoring using serial complement fixation (CF) titers in serum and cerebrospinal fluid (CSF), and neurosurgical interventions, such as ventriculoperitoneal shunt placement, in cases complicated by hydrocephalus [12,14]. Despite advances in therapy, children with CM remain at risk for severe complications, including hydrocephalus, vascular cerebritis or stroke, brain abscess, spinal involvement and, more rarely, isolated fourth ventricle (IFV) [15,16].

A significant milestone occurred in 2000 when Galgiani et al. published updated CM treatment guidelines [17], establishing fluconazole as the first-line therapy, supported by evidence from a 1993 clinical trial and the NIAID-Mycosis Study Group, which included pediatric patients [14]. Subsequent guidelines (IDSA 2016 [18]; Johnson et al. 2018 [12]) refined management standards but did not specifically address pediatric patients (>1 month and ≤19 years) as a distinct group.

Despite advances in antifungal therapy, the clinical course, risk factors, and outcomes of pediatric CM remain poorly characterized, particularly in the fluconazole era. To address this gap, we conducted a systematic review and proportional synthesis of cases reported between 2000 and 2025, when fluconazole became the established first-line therapy [17]. Our objectives were to delineate diagnostic approaches, treatment strategies, CNS complications, and outcomes in children with CM. Ultimately, our objective is to enhance clinical awareness and support evidence-based pediatric management strategies for this rare but devastating infection.

## 2. Materials and Methods

A systematic literature search was conducted in PubMed/MEDLINE, Embase (Ovid), and Web of Science from 1 January 2000 to 1 June 2025, following PRISMA 2020 guidelines. Keywords and their combinations included: “coccidioidal meningitis,” “central nervous system coccidioidomycosis,” “neurococcidioidomycosis,” “fungal meningitis,” and “coccidioidomycosis.” Extracted data included demographics, diagnostic methods, MRI findings, neurological complications, pharmacological treatments, surgical management, and patient outcomes. PROSPERO registration ID (1130290).

Inclusion criteria included articles published in English, case reports or series describing pediatric patients aged > 1 month and ≤19 years with CNS involvement due to coccidioidomycosis, and availability of clinical, diagnostic methods, treatment, CNS complications (e.g., hydrocephalus), and/or outcome data. Exclusion criteria included reviews, editorials, conference abstracts, studies without primary patient-level data, reports involving exclusively adult populations, animal studies or basic science investigations, cases diagnosed or treated before 2000, neonates (<1 month), congenital infections, pregnant patients, occupational or prison cohorts, and studies without full-text availability.

Titles and abstracts were screened to remove irrelevant studies, followed by full-text review. MFDLV, PND, and JACR manually analyzed a total of 137 manuscripts; no AI tools or programs were used. Epidemiologic studies or case series lacking at least one confirmed pediatric CM/CNS case (via CF, culture, PCR, pathological findings, or autopsy) were excluded (*n* = 105). Figure 1 illustrates the search process.

Records were screened and excluded following the initial database search in accordance with the PRISMA 2020 guidelines. Title, abstract, and full-text review were performed manually by the authors without the use of AI tools, applying predefined exclusion criteria*.

* Exclusion criteria: reported adult cases (over 19 years old) or elderly patients; pediatric cases or cases lacking a description by age were not included. Neonatal cases (under 1 month), cases involving pregnant women with congenital infections in the fetus or neonates, studies on the teratogenic effects of antifungal drugs, and review articles or guidelines on diagnosis and treatment that contained no reported cases were excluded. Additionally, prison and inmate populations, as well as workers, employees, and civilians such as construction crews, military personnel, and archaeologists, were excluded, along with diagnoses other than coccidioidomycosis (e.g., tuberculosis, Blastomyces dermatitidis).

** Inclusion criteria: Epidemiological studies and/or case series that describe at least one pediatric case of coccidioidal meningitis or CNS involvement confirmed through a diagnostic method. Source: Page MJ, et al. BMJ 2021;372:n71. doi:10.1136/bmj.n71. This work is licensed under CC BY 4.0. To view a copy of this license, visit https://creativecommons.org/licenses/by/4.0/ accessed on 26 August 2025.

Extracted data were organized into summary tables to improve clarity and interpretability. Descriptive statistics were performed on demographic and clinical variables. Bivariate analyses (chi-squared tests) were conducted for cases with complete data to explore associations between potential risk factors—including age, sex, ethnicity, immunodeficiencies, prior pulmonary disease, diagnostic methods, treatment, CNS complications, follow-up, and outcomes in CM presentation. Binary logistic regression was subsequently performed to identify factors associated with adverse outcomes. All statistical analyses were conducted using IBM SPSS Statistics v30.0 (IBM Corp., Armonk, NY, USA), with significance set at *p* < 0.05. The Joanna Briggs Institute (JBI) critical appraisal tools (Joanna Briggs Institute, Adelaide, Australia) were applied to assess risk of bias [19].

## 3. Results

Among 1089 studies, 223 were screened, and 31 met the inclusion criteria, encompassing 3874 pediatric cases. CM/CNS involvement was confirmed in 165 children (4.25%; 95% CI: 3.6–4.9%). Hydrocephalus occurred in 62 cases (37.5%; 95% CI: 30.1–44.9%), with three cases involving an isolated fourth ventricle (1.8%; 95% CI: 0.4–5.2%) (see Appendix A, Appendix A).

### 3.1. Demographics and Immunodeficiency

Among 48 pediatric cases with individual data (see Appendix A, Appendix A), the mean age at diagnosis was 105.1 months (SD = 75.6; range, 2–228 months), with 37.5% of the cases aged ≤ 5 years. Gender distribution was equal, and 48% identified as Hispanic. Immunodeficiency was present in 37.5% of cases, most commonly hyper-IgE syndrome (HIES)/Job syndrome (28%), followed by STAT1/STAT3 mutations, juvenile idiopathic arthritis, asthma, DAVID syndrome, diffuse proliferative glomerulonephritis, and ALL. HIV testing (10 cases) was negative. Immunodeficiency was more frequent in females (76.5% vs. 23.5%, *p* = 0.004) and associated with abnormal chest X-ray (CXR) or prior pneumonia (*p* = 0.030) (see Table 1).

Neurological complications were more common in immunodeficient patients: hydrocephalus alone (44%, *p* = 0.002), and hydrocephalus plus cerebral vasculitis (61%, *p* = 0.017).

### 3.2. Diagnostics

Serologic testing was positive in 79% (38/48) of cases, with serum CF reactivity in 73% (35/48). The median serum CF titer at diagnosis was 1:16 (range, 1:2–1:2048); 58% of patients had titers ≥ 1:16, and 62% reached titers ≥ 1:32. CSF CF testing was positive in 44% of cases, with titers ≥ 1:2 considered diagnostic. Culture positivity was documented in 42%, CNS involvement was confirmed in 65%, and histopathological findings in 23% (Table 2).

Among patients with hydrocephalus, 71% initially presented with CF serum titers ≥ 1:16 and/or CSF CF titers ≥ 1:2. A positive CSF CF (≥1:2) was detected in 60% of cases (*p* = 0.008). In contrast, culture was positive in only 29% (*p* = 0.007), and biopsy or cytology in 15% (*p* = 0.035). In three patients, CM was diagnosed by CSF PCR.

In cases of hydrocephalus plus stroke, 68% initially presented with CF serum titers ≥ 1:16 and/or CSF CF titers ≥ 1:2. Serum CF ≥ 1:16 was observed in 24% (*p* = 0.027), and serum CF ≥ 1:32 as the highest recorded titer in 24% (*p* = 0.006). CSF CF positivity (≥1:2) was identified in 55% (*p* = 0.002). Culture was positive in 34% (*p* = 0.041), and biopsy/cytology in 16% (*p* = 0.022).

Among patients with adverse outcomes (relapse, progression, or death), 57% presented with CF serum titers ≥ 1:16 at diagnosis (*p* = 0.013), and 64% with CF serum titers ≥ 1:32 at the highest level (*p* = 0.004). Only 14% demonstrated CSF CF positivity (≥1:2, *p* = 0.008).

### 3.3. CNS Complications

Hydrocephalus was the most frequent complication, occurring in 71% of cases and frequently requiring neurosurgical intervention. Hydrocephalus combined with stroke or vasculitis was observed in 79%. Presentation with hydrocephalus was significantly associated with age > 5 years (53%, *p* = 0.033), Hispanic ethnicity (62%, *p* = 0.011), immunodeficiency (24%, *p* = 0.002), and abnormal chest imaging or a prior history of pneumonia (18%, *p* = 0.002). Among patients with hydrocephalus plus stroke, 21% developed adverse outcomes (relapse, progression, or death) (*p* = 0.016), with an overall mortality rate of 16% (*p* = 0.027).

Less common complications included spinal involvement (19%), brain abscess (13%), and isolated fourth ventricle (IFV) (6%, *n =* 3).

### 3.4. Medical Treatment

Fluconazole was the predominant first-line therapy in this cohort, administered to 65% of patients (31/48). It was used as monotherapy in 20 cases and in combination with intravenous (IV) amphotericin B (n = 10) or itraconazole (n = 1). Other azoles, including voriconazole (n = 4) and itraconazole (n = 2), were prescribed less frequently (Table 3).

Fluconazole dosing was in the range 10–16 mg/kg/day (200–800 mg/day) or 4.5–7 mg/kg/day orally, consistent with standard pediatric regimens. Among patients who developed hydrocephalus, 68% (23/34) were initially treated with fluconazole, with 74% (17/23) receiving it as monotherapy. Similarly, 66% (25/38) of patients with both hydrocephalus and cerebral stroke were treated with fluconazole. Overall, 77% of patients receiving fluconazole achieved stable disease or survival, while 23% progressed (*p* = 0.175).

Liposomal amphotericin B (L-AmB) was used as first-line monotherapy in 10 patients and in combination with azoles in 14 patients, at IV doses of 2.5–5 mg/kg/day. Intrathecal deoxycholate amphotericin B (IT AmB-D) was administered in various regimens, including 0.05–0.3 mg/week, 1 mg/kg/day, or 30 mg/day. Dexamethasone was co-administered in some cases at 0.3 mg/kg/day.

Lifelong azole maintenance therapy was reported in 54% (26/48) of cases, primarily with fluconazole. This strategy was more common in patients with hydrocephalus (68%, *p* = 0.003) and in those with hydrocephalus plus cerebral stroke (66%, *p* = 0.002). Lifelong therapy was associated with improved outcomes, with 85% of patients achieving stable disease compared to 16% who experienced relapse, progression, or death (*p* = 0.022).

### 3.5. Neurosurgical Management

Neurosurgical interventions were performed in 65% of patients (31/48), with an average of 1.7 procedures per case (SD = 2.7; range 0–15). The majority of procedures (94%, 29/31) were derivative shunts for the management of hydrocephalus, including ventriculoperitoneal shunts, external ventricular drains, and Ommaya reservoirs. Additional interventions included one third ventriculostomy, two fourth ventricle shuntings, one surgical drainage of a recurrent subgaleal abscess, and meningeal biopsies. The median number of shunt-related surgeries was two (range 1–15). Overall, 55% of patients required two or more procedures, while 45% underwent only one.

The frequency of re-intervention varied across subgroups. Among patients with hydrocephalus, 44% required two or more surgeries, 41% had one, and 15% had none (*p* < 0.001). A similar pattern was observed in patients with hydrocephalus and cerebral stroke, where 40% underwent two or more surgeries, 37% one, and 23% none (*p* = 0.003). Positive CSF CF (titers ≥ 1:2) were also associated with higher surgical requirements, as 38% underwent two or more interventions, 48% one, and 14% none (*p* = 0.007).

The type of antifungal regimen had a significant impact on surgical burden. Among patients receiving fluconazole as initial monotherapy (*n =* 20), 45% required ≥2 surgeries, 40% underwent a single procedure, and 15% required none (*p* = 0.034). In contrast, 73% of patients treated with fluconazole in combination with other therapies did not require any surgical intervention (*p* = 0.034). Patients receiving monotherapy as subsequent therapy had a higher surgical burden (52%) compared with those on dual (9%) or triple therapy (0%) (*p* = 0.017). Even among patients on lifelong azole therapy, 50% required ≥2 surgeries, 39% required one, and 12% required none (*p* < 0.001).

### 3.6. Follow-Up and Outcomes

Follow-up was available in 42 cases, average time 23 months (SD = 27.8), range 0.3–108 months. Overall mortality was 19% (9/48), while 71% were alive/disease control, and 10% experienced relapse or disease progression. Serum CF titers ≥ 1:16 were associated with adverse outcomes (53% relapse/death, *p* = 0.019; mortality 27%). Titers ≥ 1:32 showed a similar risk (56% events, *p* = 0.003; mortality, 25%). In contrast, CSF CF ≥1:2 was associated with favorable outcomes, with 91% survival and 9 adverse events (*p* = 0.008, 5% mortality). Patients diagnosed by biopsy/cytology had higher rates of relapse or death (55%, *p* = 0.035).

Mortality among patients with hydrocephalus was 18%, and 16% in those with hydrocephalus plus stroke. Hydrocephalus alone did not show a significant difference in outcomes, but cases with hydrocephalus plus cerebral stroke showed significant differences in survival rates (21% presented with relapse, recurrence, or death, *p* = 0.016).

### 3.7. Long-Term Therapy and Prognostic Factors

Lifelong azole therapy significantly improved outcomes, with an 85% survival rate versus a 15% relapse/death rate (*p* = 0.022). Kaplan–Meier analysis confirmed a more prolonged overall survival (mean, 82 vs. 32 months; *p* = 0.002).

Logistic regression identified serum CF titers ≥ 1:16 as an independent predictor of adverse events (OR = 4.5, 95% CI 1.097–18.465, *p* = 0.037, Table 4). At the same time, age, immunosuppression, hydrocephalus, and fluconazole as first-line treatment were not significant (Appendix A, Appendix A).

## 4. Discussion

Coccidioidal meningitis (CM) represents the most severe form of extrapulmonary coccidioidomycosis, occurring in up to 16% of disseminated cases [5]. Pediatric CM remains rare, affecting approximately 1% of children, despite pediatric cases accounting for 5–10% of all infections [6,7]. Untreated, CM is often fatal within one year [8]. This systematic review examined pediatric CM cases reported during the fluconazole era (2000–2025) [17]. Given the lack of pediatric-specific guidelines for CM [12,13,18], this systematic review aimed to clarify diagnostic approaches, treatment strategies, CNS complications, and outcomes, highlighting the urgent need for evidence-based management protocols in this population.

In our analysis, CM was reported in 4.25% of children with coccidioidomycosis, exceeding previous literature estimates of approximately 1% in pediatric populations [6,7]. Hydrocephalus occurred in about 38% of CM cases, with three cases presenting an isolated fourth ventricle (IFV). Although these findings likely underestimate the true incidence of pediatric CM, many cases of meningitis of unknown etiology may correspond to this infection [20]. Notably, diagnostic categorization of CM has predominantly relied on clinicopathological studies, as outlined in the proposed clinicopathological categorization system for clinical research in coccidioidomycosis [21]. Additionally, the identification of pediatric CM cases in Egypt and China underscores the global relevance of this disease and highlights the lack of standardized diagnostic and treatment protocols outside the Americas [3,4].

### 4.1. Diagnosis

Prompt inclusion of CM in the differential diagnosis is essential for children with neurological symptoms in endemic regions. Confirmation relies on neuroimaging, serology, CSF analysis, culture, and histopathological findings [12]. Serologic testing remains a cornerstone due to the challenges of isolating *Coccidioides* spp., with up to 80% of cases showing antibody positivity via immunodiffusion (ID), complement fixation (CF), or enzyme immunoassay (EIA) [22,23,24,25,26]. In our cohort, 79% had positive serology, with 73% showing serum CF reactivity (median 1:16); 58% of cases presented with serum titers ≥ 1:16. CSF CF positivity was lower (44%), but titers ≥ 1:2 were diagnostic, supporting the importance of testing both serum and CSF. Culture and histopathological findings, although highly specific, had lower sensitivity (42% and 23%, respectively), in line with previous reports [12]. Importantly, PCR provided additional diagnostic value in three cases, supporting its role as a complementary tool in challenging presentations.

In our results, CSF CF titers ≥ 1:2 were significantly associated with hydrocephalus (60%, *p* = 0.008), whereas serum CF titers ≥ 1:16, although standard (71%), were not statistically significant. In cases with hydrocephalus plus cerebral stroke, combined serum and CSF CF testing improved diagnostic accuracy, with serum CF ≥ 1:16 or CSF CF ≥ 1:2 found in 68% of cases, and 24% exhibited serum CF ≥ 1:16 (*p* = 0.027). Culture and biopsy had a limited yield (≤34%). Importantly, adverse outcomes were associated with higher serum CF titers: 57% had titers ≥ 1:16 at diagnosis (*p* = 0.013), and 64% reached titers ≥ 1:32 at peak (*p* = 0.004), highlighting the prognostic value of serum CF monitoring.

### 4.2. Neurological Complications

Neuroimaging, particularly gadolinium-enhanced MRI, is critical for the diagnosis and prognosis of CM [27,28]. Around half of patients with CNS involvement show radiological abnormalities, most frequently hydrocephalus (40–51.6%), basal meningitis (46.8%), and cerebral infarcts (38.7%) [29]. Hydrocephalus was the predominant CNS complication in our results (71%), often requiring surgical intervention. Importantly, hydrocephalus combined with cerebral stroke or vasculitis represented 79% of complicated cases and was significantly associated with relapse and mortality (21%, *p* = 0.016). Less common complications included spinal involvement (19%), brain abscesses (13%), and IFV (6%), further highlighting disease severity. High initial serum CF titers (≥1:16) and immunodeficiency are key predictors of CNS complications, primarily hydrocephalus combined with cerebral stroke (*p* < 0.05). These findings underscore the importance of early diagnosis, close neuroimaging monitoring, and aggressive management to optimize outcomes in this vulnerable population.

### 4.3. Medical Treatment

Fluconazole remains the mainstay of therapy for pediatric coccidioidal meningitis due to its oral availability, CNS penetration, and favorable safety profile [12,18]. In our study, 65% of cases received fluconazole as the first-line therapy, with doses ranging from 10 to 16 mg/kg/day (200 to 800 mg/day). Lifelong azole therapy was associated with significantly better outcomes, including higher rates of stable disease (85% versus 15% with relapse or death, *p* = 0.022) and longer survival (mean OS of 82 vs. 32 months; *p* = 0.002). High initial serum CF titers (≥1:16) independently predicted adverse outcomes (OR = 4.5, *p* = 0.037), underscoring their value for early risk stratification. These findings highlight the importance of identifying high-risk patients and maintaining adherence to lifelong azole therapy to improve prognosis in pediatric CM.

### 4.4. Neurosurgical Procedures

Neurosurgical intervention is a critical component in managing pediatric coccidioidal meningitis, particularly in patients with hydrocephalus, which often requires shunt placement due to CSF obstruction from meningeal inflammation [12,18,30]. Rare complications, such as isolated fourth ventricle syndrome, may necessitate advanced procedures, including endoscopic aqueductoplasty [18,31]. In our cohort, 65% of patients underwent surgical intervention, with many requiring multiple procedures due to shunt failure or complications, including Ommaya reservoir placement after first-line therapy failure. Re-intervention occurred in 55% of cases and was more frequent among patients with hydrocephalus (44%, *p* < 0.001), hydrocephalus with cerebral infarction (40%, *p* = 0.003), or positive CSF CF titers (≥1:2) (38%, *p* = 0.007). Surgical burden was influenced by antifungal therapy: fluconazole monotherapy as first-line treatment, as well as subsequent monotherapy (with azoles or AmB), was associated with higher rates of re-intervention (initial: 45%, *p* = 0.034; subsequent: 52%, *p* = 0.017), whereas combination or lifelong azole therapy reduced—but did not eliminate—the need for repeat surgery. These findings underscore the importance of early diagnosis, aggressive antifungal therapy, and strategic neurosurgical planning for achieving optimal outcomes in pediatric CM, with treatment adherence and complete follow-up remaining crucial determinants of prognosis.

### 4.5. Follow-Up and Outcomes

This study highlights the critical role of early recognition and sustained antifungal therapy in pediatric coccidioidal meningitis. In our cohort, 65% (31/48) of patients received fluconazole as first-line therapy, with 20 cases treated as monotherapy, and 54% (26/48) maintained on lifelong azole therapy. Overall mortality was 19%, markedly lower than the nearly 100% mortality reported in untreated cases, underscoring the lifesaving impact of timely intervention. Hydrocephalus combined with cerebral infarction was associated with the presence of adverse outcomes (relapse or death), while lifelong azole therapy was significantly linked to improved survival. Conversely, high serum CF titers (≥1:16) were associated with poor outcomes and were identified as an independent risk factor in this study. These findings underscore the importance of early diagnosis, personalized antifungal regimens, and meticulous neuroimaging-based risk stratification to optimize prognosis in pediatric CM.

### 4.6. Limitations

This study has several limitations. It relies on case reports, case series, and retrospective studies, which are inherently prone to reporting bias, variability in data quality, and a lack of standardized protocols. To ensure methodological rigor, we applied the Critical Appraisal Tools for use in JBI Systematic Reviews [19]. Due to heterogeneity and incomplete reporting, only cases with complete individual data (*n =* 48) were included, underscoring the importance of developing standardized guidelines for pediatric CM. Missing or incomplete data—such as unclear diagnostic timelines, length of stay, initial misdiagnosis, follow-up durations, and treatment outcomes—may influence interpretation. Histopathological data were limited, with only a few patients showing cytologic evidence of Coccidioides and meningeal inflammation [16]. Of three PCR-diagnosed cases, only one specified the molecular target, using the Ag2/PRA gene and 28S rDNA [32]; the other two [16,33] confirmed PCR in CSF without details on the gene or sequence. Publication bias may have inflated the frequency of complications such as hydrocephalus or adverse outcomes. Most studies originate from endemic regions, which limits their generalizability. Despite these limitations, this review provides valuable insights into pediatric CM and highlights the need for prospective, multicenter studies to inform evidence-based management strategies.

## 5. Conclusions

Pediatric coccidioidal meningitis (CM) is a rare but life-threatening condition associated with substantial CNS morbidity. Hydrocephalus is the most frequent and severe complication, and its combination with cerebral vasculitis appears to influence patient outcomes adversely. High initial serum CF titers (≥1:16) are essential predictors of CNS complications, particularly hydrocephalus with stroke, and are significantly associated with worse outcomes. Lifelong azole therapy yields significant long-term benefits, underscoring the importance of both early risk stratification and adherence to treatment. These findings underscore the urgent need for standardized, pediatric-specific diagnostic and management guidelines.

## Figures and Tables

**Figure 1 jof-11-00713-f001:**
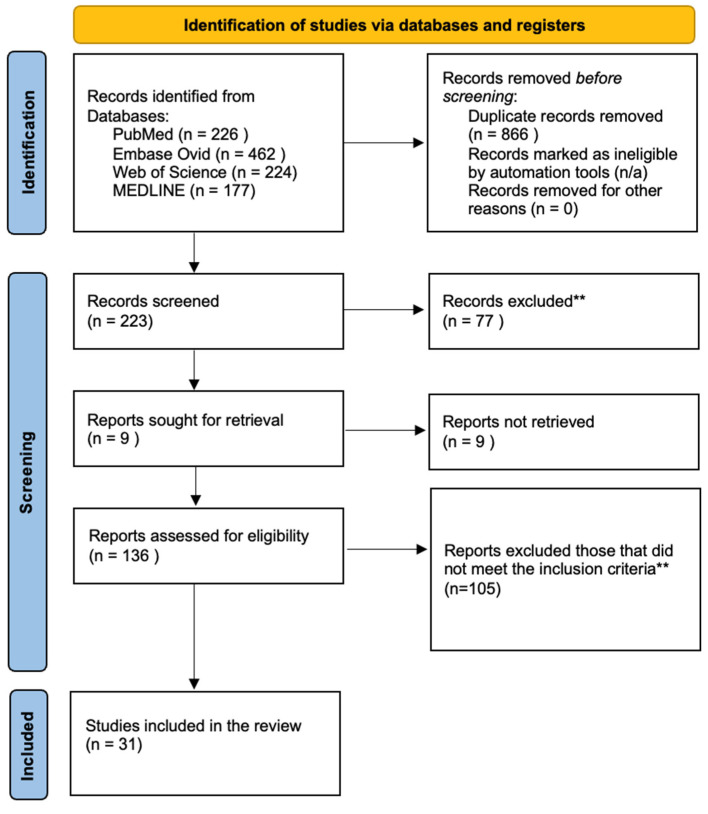
A flow diagram showing the searching process.

**Table 1 jof-11-00713-t001:** Demographic data, pediatric cases with CM in the fluconazole era (2000–2025) (total *n* = 48).

Variable	*n* (%)
Age (months)	mean 105.1 SD = 75.6
Sex	
Male	24 (50)
Female	23 (48)
Unknown	1 (2)
Ethnicity	
Hispanic/Latinx	23 (48)
White (no Hispanic)	9 (19)
Black or African American	4 (8)
Asian	3 (6)
Other/unknown	9 (19)
Immunodeficiency (yes)	18 (37.5)
No/unknown	30 (62.5)
CXR normal/negative	6 (12.5)
CRX abnormal/prior pneumonia	15 (31.3)
Unknown	27 (56.3)
CNS complications (Neuroimaging or autopsy)	
Hydrocephalus	34 (70.8)
Cerebral vasculitis/stroke	10 (20.8)
Brain/cerebellar abscess	6 (12.5)
Spinal alterations	9 (18.8)
IFV	3 (6.3)
Hydrocephalus + stroke	38 (79.2)
Status	
Alive/stable	34 (71)
Relapse/progressive diseases	5 (10)
Died	9 (19)

CM = coccidioidal meningitis; CNS = central nervous system; CXR = chest X-ray (radiograph); IFV = isolated fourth ventricle, SD = Standard deviation.

**Table 2 jof-11-00713-t002:** Diagnostic methods for pediatric cases with CM (*n* = 48).

Method	*n* (%)
Serologies (CF, ID, EIA)	
Positive	38 (79)
Negative/unknown	10 (21)
CF serum	
Positive	21 (44)
Negative/unknown	27 (56)
CF serum (initial diagnostic level)	Median 1:16 (range 1:2 to 1:2048)
CF serum (highest level)	Median 1:32 (range 1:2 to 1:2048)
CSF CF	
Positive	21 (44)
Negative/unknown	27 (56)
CSF CF (level initial diagnostic)	1:2 (range 1:2 to 1:32)
Culture	
Positive	20 (42)
Negative/unknown	28 (58)
Positive cultures	(*n =* 20)
Cultures (CNS or CSF)	13 (65)
Other sites	7 (35)
Histopathological findings/cytology	
Positive	11 (23)
Negative/unknowns	37 (77)
RT-PCR/DNA	
Positive	3 (6)
Negative/unknown	45 (94)

CF = complement fixation; CNS = central nervous system; CSF = cerebrospinal fluid; DNA = deoxyribonucleic acid; EIA = enzyme immunoassay; ID = immunodiffusion; RT-PCR = reverse transcription polymerase chain reaction.

**Table 3 jof-11-00713-t003:** Medical treatment reported in pediatric cases with CM (total *n =* 48).

Therapy	*n* (%)
Initial antifungal therapy	
FLU	31 (65)
Monotherapy	20 (65)
Combination therapy	11 (35)
Other therapy	17 (35)
AmB IT or IV	10 (37)
Itraconazole	1 (6)
Voriconazole	1 (6)
L-AmB + voriconazole	4 (24)
Patient died before treatment	1 (6)
Maintenance therapy	
L-AmB monotherapy	10 (21)
Azole monotherapy	21 (44)
AmB + azole	8 (17)
V/C	3 (7)
Triple therapy/other combinations	5 (10)
Patient died before treatment	1(2)
Treatment schemas	
Monotherapy	31 (66)
Doble therapy	11 (23)
Triple therapy	5 (11)
Other therapies	
INF- γ	2 (4)
Long-term/lifelong azole therapy	26 (54)

AmB = amphotericin B, FLU = fluconazole, INF-γ = interferon-gamma, IT = intrathecal, IV = intravenous, L-AmB = liposomal amphotericin B, V/C = voriconazole plus caspofungin.

**Table 4 jof-11-00713-t004:** Logistic regression model for adverse outcomes in pediatric CM patients.

Variable	OR	95% CI Lower	95% CI Upper	*p*-Value
Serum CF titers ≥ 1:16	4.500	1.097	18.465	0.037
Lifelong azole	0.250	0.060	1.040	0.057
Constant	0.444	-	-	0.139

OR = odds ratio; CI = confidence interval; CF = complement fixation. Dependent variable: adverse outcome (progression, relapse, or death). Logistic regression was performed using the backward likelihood ratio (backward: LR) method.

## Data Availability

The original contributions presented in the study are included in the article/Appendix A, further inquiries can be directed to the corresponding author.

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
