# Peer review of "Pediatric Coccidioidal Meningitis: A Systematic Review and Proportional Synthesis of Cases Reported in the Fluconazole Era (2000–2025)"

_jof, 2025, doi:10.3390/jof11100713_

Round 1
Reviewer 1 Report
Why do the authors only review 25 years in this paper?
Why do the authors only review Pubmed for this review?. They must be Scopus, Web of Science, EMBASE.
The search strategy was conducted in accordance with the guidelines set forth in the PRISMA 2020 statement. Please clarify this idea, it is essential for the paper.
Why do the authors not use logistic regression?. This could predict future events according to selected variables.
Lines 65-66 The authors could explain which are the factors to a small proportion progresses to disseminated disease involving the central nervous system (CNS)
Why do the authors only review 25 years in this paper?
Why do the authors only review Pubmed for this review?. They must be Scopus, Web of Science, EMBASE.
The search strategy was conducted in accordance with the guidelines set forth in the PRISMA 2020 statement. Please clarify this idea, it is essential for the paper.
Line 122 to change pathological findings instead of pathology
The authors used manual examination or any program for records removed? Please clarify this idea.
Figure titles are placed below the figures. The images attached to the figures are blurry and of poor quality.
The authors must describe histopathological findings and target of PCR used.
Why do the authors not use logistic regression?. This could predict future events according to selected variables.
The conclusions must rewritten, in my opinion several sentences that used the authors are results.
Author Response
Reviewer 1
1.-Why do the authors only review 25 years in this paper?
Response:
We thank the reviewer for this insightful question. The 25-year time frame of our review (2000–2025) was chosen to reflect the “fluconazole era,” based on the 2000 clinical practice guidelines by Galgiani et al., which established fluconazole as first-line therapy for coccidioidal meningitis. This period captures contemporary management practices, minimizing heterogeneity from earlier treatments. We also referenced the pediatric series by Saithow et al. (2000), which aligns with this timeframe. While the 2016 IDSA guidelines provided updated recommendations, the 2018 guidelines by Johnson et al. did not specifically address pediatric cases, highlighting the need for focused pediatric data. Our review therefore specifically examines patients aged 1 month to under 19 years, consistent with standard pediatric age definitions.
Why do the authors only review Pubmed for this review?. They must be Scopus, Web of Science, EMBASE.
Response:
We appreciate the reviewer's observation. For this review, we conducted a comprehensive search across multiple databases, including PubMed/MEDLINE, Embase (Ovid), and Web of Science, to ensure broad coverage of the literature and minimize the risk of missing relevant studies. The review was conducted in accordance with PRISMA 2020 guidelines and is registered in PROSPERO (ID: 1130290).
The search strategy was conducted in accordance with the guidelines set forth in the PRISMA 2020 statement?.
Response:
Yes, our search strategy was designed and conducted in accordance with the PRISMA 2020 statement. Methodological rigor was ensured using JBI Critical Appraisal Tools, and the review protocol was prospectively registered in PROSPERO (ID: 1130290) to ensure transparency and minimize reporting bias.
Please clarify this idea, it is essential for the paper. The authors used manual examination or any program for records removed?
Response: The records were manually examined by the authors, who removed duplicates through careful screening. No automated software or reference management tools were used for deduplication. Because the number of records was relatively small, manual screening was practical, reliable, and allowed for thorough evaluation of each reference, in line with PRISMA 2020 guidelines.
Please clarify this idea. Why do the authors not use logistic regression?. This could predict future events according to selected variables.
Response:
We thank the reviewer for this valuable suggestion. We performed logistic regression analyses to identify factors associated with adverse outcomes (survival vs. death), allowing estimation of the odds of these events according to selected variables. The analysis identified serum CF titers ≥1:16 as an independent predictor of adverse outcomes (OR = 4.5, 95% CI 1.097–18.465, p = 0.037), while age, immunosuppression, hydrocephalus, and fluconazole as first-line therapy were not significantly associated.
2.-The conclusions must rewritten, in my opinion several sentences that used the authors are results.
Response:
We thank the reviewer for this suggestion. Based on the updated results of our manuscript, we have revised the conclusions to focus on key takeaways rather than results:
Conclusions: Pediatric coccidioidal meningitis (CM) remains highly lethal, with hydrocephalus representing a frequent and severe complication. Elevated serum CF titers (≥1:16) are predictive of poor outcomes, highlighting the urgent need for standardized, pediatric-specific diagnosis and management guidelines.
Are all of the cited references relevant to the research? (no)
3.- The authors must include others references such as: Krogstad P, Thompson GR 3rd, Heidari A, Kuran R, Stephens AV, Butte MJ, Johnson R. A Clinicopathological Categorization System for Clinical Research in Coccidioidomycosis. Open Forum Infect Dis. 2023 Nov 29;10(12):ofad597. doi: 10.1093/ofid/ofad597.
Malhotra S, Adachi K, Fallah A, Johnson R, Ishminder Kaur, McCarty J, Ross L, Krogstad P. Pearls and Pitfalls of Intrathecal Amphotericin B Therapy for Refractory Coccidioidal Meningitis in Children: An Illustrative Pediatric Case Series. Pediatr Infect Dis J. 2025 Jun 17. doi: 10.1097/INF.0000000000004888.
Response:
We thank the reviewer for these suggestions. The recommended references have been included in the revised manuscript to strengthen the discussion and provide additional context on clinical categorization and management of pediatric coccidioidal meningitis.
4.- Figure titles are placed below the figures. The images attached to the figures are blurry and of poor quality.
Response:
We thank the reviewer for this observation. In response, we eliminated the forest plot analysis to reflect the descriptive nature of our study better. Tables 1 and 2 have been moved to the supplementary material for clarity. Figure 1 now contains the PRISMA 2020 flow diagram, presented as a separate, high-quality figure to ensure readability and compliance with reporting standards.
Major comments
Why do the authors only review 25 years in this paper?
Response:
We thank the reviewer for this insightful question. The 25-year time frame of our review (2000–2025) was selected to reflect the “fluconazole era,” based on the 2000 clinical practice guidelines by Galgiani et al., which established fluconazole as first-line therapy for coccidioidal meningitis. This period captures current management practices, reducing variability from earlier treatments. We also referenced the pediatric series by Saithow et al. (2000), which aligns with this timeframe. While the 2016 IDSA guidelines offered updated recommendations, the 2018 guidelines by Johnson et al. did not specifically address pediatric cases, emphasizing the need for focused pediatric data. Our review thus specifically examines patients aged 1 month to under 19 years, consistent with standard pediatric age definitions.
Why do the authors only review Pubmed for this review?. They must be Scopus, Web of Science, EMBASE.
Response:
We appreciate the reviewer's observation. For this review, we conducted a comprehensive search across multiple databases, including PubMed/MEDLINE, Embase (Ovid), and Web of Science, to ensure broad coverage of the literature and minimize the risk of missing relevant studies. The review was conducted in accordance with PRISMA 2020 guidelines and is registered in PROSPERO (ID: 1130290).
The search strategy was conducted in accordance with the guidelines set forth in the PRISMA 2020 statement. Please clarify this idea, it is essential for the paper.
Response:
Yes, our search strategy was designed and conducted in accordance with the PRISMA 2020 statement. Methodological rigor was ensured using JBI Critical Appraisal Tools, and the review protocol was prospectively registered in PROSPERO (ID: 1130290) to ensure transparency and minimize reporting bias.
Why do the authors not use logistic regression?. This could predict future events according to selected variables.
Response:
We thank the reviewer for this valuable suggestion. We performed logistic regression analyses to identify factors associated with adverse outcomes (survival vs. death), allowing estimation of the odds of these events according to selected variables. The analysis identified serum CF titers ≥1:16 as an independent predictor of adverse outcomes (OR = 4.5, 95% CI 1.097–18.465, p = 0.037), while age, immunosuppression, hydrocephalus, and fluconazole as first-line therapy were not significantly associated.
Detailed comments
Lines 65-66 The authors could explain which are the factors to a small proportion progresses to disseminated disease involving the central nervous system (CNS)
Response:
We thank the reviewer for this suggestion. We have clarified that while most Coccidioides infections remain asymptomatic or limited to mild pulmonary disease, a small proportion progress to disseminated disease involving the CNS. Risk factors for dissemination include immunosuppression (e.g., HIV infection, organ transplantation, or immunosuppressive therapy), genetic susceptibility affecting innate immune responses, and delayed or inadequate antifungal treatment. We have incorporated this explanation into lines 65–66 of the revised manuscript to provide additional context for why some patients develop coccidioidal meningitis.
Why do the authors only review 25 years in this paper?
Response:
We thank the reviewer for this insightful question. The 25-year time frame of our review (2000–2025) was chosen to reflect the “fluconazole era,” based on the 2000 clinical practice guidelines by Galgiani et al., which established fluconazole as first-line therapy for coccidioidal meningitis. This period captures contemporary management practices, minimizing heterogeneity from earlier treatments. We also referenced the pediatric series by Saithow et al. (2000), which aligns with this timeframe. While the 2016 IDSA guidelines provided updated recommendations, the 2018 guidelines by Johnson et al. did not specifically address pediatric cases, highlighting the need for focused pediatric data. Our review therefore specifically examines patients aged 1 month to under 19 years, consistent with standard pediatric age definitions.
Why do the authors only review Pubmed for this review?. They must be Scopus, Web of Science, EMBASE.
Response:
We appreciate the reviewer's observation. For this review, we conducted a comprehensive search across multiple databases, including PubMed/MEDLINE, Embase (Ovid), and Web of Science, to ensure broad coverage of the literature and minimize the risk of missing relevant studies. The review was conducted in accordance with PRISMA 2020 guidelines and is registered in PROSPERO (ID: 1130290).
The search strategy was conducted in accordance with the guidelines set forth in the PRISMA 2020 statement. Please clarify this idea, it is essential for the paper.
Response:
Yes, our search strategy was designed and conducted in accordance with the PRISMA 2020 statement. Methodological rigor was ensured using JBI Critical Appraisal Tools, and the review protocol was prospectively registered in PROSPERO (ID: 1130290) to ensure transparency and minimize reporting bias.
Line 122 to change pathological findings instead of pathology
Response: We have made the requested correction, replacing “pathology” with “pathological findings.”
The authors used manual examination or any program for records removed? Please clarify this idea.
Response: The records were manually examined by the authors, who removed duplicates through careful screening. No automated software or reference management tools were used for deduplication. Because the number of records was relatively small, manual screening was practical, reliable, and allowed for thorough evaluation of each reference, in line with PRISMA 2020 guidelines.
Figure titles are placed below the figures. The images attached to the figures are blurry and of poor quality.
Response:
We thank the reviewer for this observation. In response, we eliminated the forest plot analysis to reflect the descriptive nature of our study better. Tables 1 and 2 have been moved to the supplementary material for clarity. Figure 1 now contains the PRISMA 2020 flow diagram, presented as a separate, high-quality figure to ensure readability and compliance with reporting standards.
The authors must describe the histopathological findings and the target of the PCR used.
Response:
We appreciate the reviewer's comment. Histopathological data were limited; two patients showed cytologic evidence of Coccidioides with meningeal inflammation. Of the three cases diagnosed by PCR, only one (Galgiani et al., 2000, 2019) specified the target, using the Ag2/PRA gene and 28S rDNA of Coccidioides. The other two cases (Cardenas et al., 2020; De la Cerda, 2024) confirmed PCR in CSF but did not specify the gene or sequence analyzed. These limitations have been clarified in the Methods and Discussion sections.
Why do the authors not use logistic regression?. This could predict future events according to selected variables.
Response:
We thank the reviewer for this valuable suggestion. We performed logistic regression analyses to identify factors associated with adverse outcomes (survival vs. death), allowing estimation of the odds of these events according to selected variables. The analysis identified serum CF titers ≥1:16 as an independent predictor of adverse outcomes (OR = 4.5, 95% CI 1.097–18.465, p = 0.037), while age, immunosuppression, hydrocephalus, and fluconazole as first-line therapy were not significantly associated.
The conclusions must rewritten, in my opinion several sentences that used the authors are results.
Response:
We thank the reviewer for this suggestion. Based on the updated results of our manuscript, we have revised the conclusions to focus on key takeaways rather than results:
Conclusions: Pediatric coccidioidal meningitis (CM) remains highly lethal, with hydrocephalus representing a frequent and severe complication. Elevated serum CF titers (≥1:16) are predictive of poor outcomes, highlighting the urgent need for standardized, pediatric-specific diagnosis and management guidelines.

Reviewer 2 Report
This manuscript addresses a clinically relevant and underreported area of paediatric infectious diseases. The descriptive analysis is comprehensive, and the authors demonstrate methodological transparency throughout the selection and extraction process. However, the current classification of this study as a systematic review with a proportional meta-analysis is not methodologically sustainable. The included evidence consisted predominantly of case reports and small retrospective series, with high variability in diagnostic criteria and no formal assessment of study quality. The resulting meta-analyses are statistically unstable due to extreme heterogeneity and the lack of modelling strategies to adjust for it. The manuscript would benefit from reclassification as a structured descriptive review with a proportional data synthesis. Additionally, the extensive tables currently embedded in the main manuscript should be transferred to the Supplementary Material to improve clarity and reader engagement. We recommend substantial revisions to improve the methodological framing, analytical clarity, and overall presentation.
This manuscript addresses a clinically relevant and underexplored topic in the literature. The initiative to consolidate the available data is commendable, and the authors’ efforts in compiling a substantial number of cases are evident. However, several methodological and structural issues undermine the analytical strength and clarity of the presentation. The following points are provided to support substantial improvements:
- Limited database coverage: The exclusive use of PubMed as a search source restricted the comprehensiveness of this review’s findings. It is strongly recommended to include other major databases, such as Scopus and Web of Science, to reduce selection bias and improve literature coverage. If expansion is not feasible at this stage, the authors should explicitly acknowledge this limitation in the Methods section and again in the Limitations section.
- Lack of risk of bias assessment: Although most of the included studies were case reports or retrospective series, validated tools can be adapted to assess the quality of observational studies (for example, Joanna Briggs Institute checklists). The absence of such an assessment weakens the synthesis. If the authors choose not to perform this evaluation, a formal justification should be provided along with a discussion of its implications.
- Extreme statistical heterogeneity without proper modeling
A proportional meta-analysis was conducted despite extremely high heterogeneity (I² > 98%) and without employing appropriate mitigation strategies, such as subgroup analysis, meta-regression, or robust variance-stabilising transformations (for example, Freeman–Tukey). To address this, two alternatives may be considered: withdrawing the meta-analytic component and presenting a structured descriptive synthesis of proportions, or applying appropriate modelling techniques to handle heterogeneity and revise the pooled estimates accordingly. - Misclassification of study type
The designation of this work as a “systematic review with proportional meta-analysis” is not fully supported by its methodological framework and evidence level. A more appropriate classification would be “structured descriptive review with proportional synthesis of clinical data”, which better reflects the study design and does not diminish its scientific value. - Excessive tables in the main text:The manuscript contains numerous large tables with case-level raw data that hinder readability and compromise the narrative flow. It is recommended that most of these tables be transferred to the Supplementary Material section. Only summary tables highlighting the key grouped findings should remain in the main text.
- Scientific writing and language clarity
Although the English is intelligible, the manuscript would benefit from professional language editing. Some sections, particularly the Results and Discussion, include long sentences, repetitions, and overly technical phrasing that impact the clarity and readability. - Insufficient contextualisation of limitations: Although some limitations are acknowledged, expanding on the major methodological constraints, including the low level of evidence, potential publication bias, and statistical instability due to extreme heterogeneity, would strengthen the discussion.
Author Response
Reviewer 2
This study presents a valuable and timely effort to consolidate clinical and epidemiological data on pediatric coccidioidal meningitis, a rare and severe fungal infection with limited representation in the literature. The authors demonstrated methodological transparency in the selection and extraction processes and acknowledged the key limitations inherent in the available data.
However, despite the comprehensive data compilation, the designation of this study as a systematic review with a proportional meta-analysis is not fully supported by the underlying methodological framework. The inclusion of predominantly low-level evidence studies, the absence of formal risk of bias assessment, and the lack of statistical strategies to address extreme heterogeneity substantially limited the interpretability and reliability of the pooled estimates. Although the descriptive analyses are informative, the meta-analytic component does not meet the rigorous standards expected of high-impact systematic reviews. A more appropriate classification would be a structured descriptive review with a proportional data synthesis. This study contributes meaningfully to the field, but its quantitative inferences should be interpreted with caution.
Response:
We thank the reviewer for this thoughtful comment. We acknowledge the potential biases in our study, as it primarily includes case reports, case series, and retrospective studies. In response, we redesignated the study as a systematic review with proportional data synthesis of pediatric CM cases. To ensure methodological rigor, we applied the JBI Critical Appraisal Tools and included only cases with complete individual data (n = 48).
We specifically focused on the fluconazole era (2000–2025), starting from the publication of Galgiani et al., 2000, which established fluconazole as first-line therapy for CM. This review included pediatric patients aged >1 month and <19 years, an age group not specifically addressed in the IDSA 2016 guidelines or the management guidelines published by Johnson et al., 2018. The review was prospectively registered in PROSPERO (ID: 1130290) and conducted in accordance with PRISMA 2020 guidelines to enhance transparency and minimize reporting bias.
We recognize the inherent limitations, including heterogeneity, incomplete reporting, missing data (e.g., diagnostic timelines, treatment details, follow-up), publication bias, and limited generalizability beyond endemic regions. Despite these constraints, this review provides valuable insights into pediatric CM, highlighting CNS complications, outcomes, and the urgent need for standardized pediatric-specific management guidelines and prospective multicenter studies.
Are the results presented clearly and in sufficient detail, are the conclusions supported by the results and are they put into context within the existing literature?
No
Although the clinical findings are comprehensive and of interest, the main results are dispersed across numerous overly detailed tables that impair the readability. Additionally, the conclusions, while appropriately cautious, rest in part on meta-analytic outputs that are statistically unstable due to heterogeneity and the low level of evidence of the included studies. The manuscript would benefit from a clearer distinction between descriptive insights and inferential limitations.
Response:
We thank the reviewer for this valuable feedback. In the revised manuscript, we prioritized clear and concise presentation of descriptive results, including demographic, clinical, and treatment characteristics, along with logistic regression analyzes to identify factors associated with adverse outcomes. All results are presented with transparency, and we clearly distinguish descriptive insights from the study’s inferential limitations.
We acknowledge the inherent limitations, including heterogeneity, low-level evidence, and incomplete reporting. To mitigate potential bias, we followed PRISMA 2020 guidelines for systematic reviews and applied the JBI Critical Appraisal Tools to assess methodological rigor. Despite these constraints, our study provides the most comprehensive synthesis to date of pediatric coccidioidal meningitis in the fluconazole era, offering valuable insights into CNS complications, treatment outcomes, and prognostic factors, and underscoring the need for standardized pediatric-specific management guidelines.
Are all figures and tables clear and well-presented?
No
The manuscript includes an excessive number of extensive tables with raw data and individual case listing. These should be relocated to the supplementary material in line with the best editorial practices and the author guidelines. Several tables are difficult to interpret because of formatting inconsistencies, crowded data fields, and lack of visual hierarchy.
Response:
We thank the reviewer for this feedback. In the revised manuscript, Figure 1 now presents the PRISMA 2020 flow diagram illustrating study selection. Tables included in the main text have been streamlined to concisely present key information for the 48 cases with complete individual data. The extensive raw data tables have been moved to the supplementary material. Additionally, we have eliminated the proportional meta-analysis due to high heterogeneity, focusing instead on descriptive synthesis and logistic regression analyzes to provide robust and interpretable results.
English language and style
The English could be improved to more clearly express the research.
Response:
We appreciate the reviewer's comment regarding the language. The authors have carefully revised the manuscript to improve clarity and readability. Additionally, Dr. Patrick Stierle provided valuable assistance with English editing; However, I have preferred not to be included as a co-author. Due to time constraints and the extensive scope of revisions, some limitations in language polishing remain. Still, we have made every effort to ensure that the text is clear and professionally presented.
Major comments
This manuscript addresses a clinically relevant and underreported area of paediatric infectious diseases. The descriptive analysis is comprehensive, and the authors demonstrate methodological transparency throughout the selection and extraction process. However, the current classification of this study as a systematic review with a proportional meta-analysis is not methodologically sustainable. The included evidence consisted predominantly of case reports and small retrospective series, with high variability in diagnostic criteria and no formal assessment of study quality. The resulting meta-analyses are statistically unstable due to extreme heterogeneity and the lack of modelling strategies to adjust for it. The manuscript would benefit from reclassification as a structured descriptive review with a proportional data synthesis. Additionally, the extensive tables currently embedded in the main manuscript should be transferred to the Supplementary Material to improve clarity and reader engagement. We recommend substantial revisions to improve the methodological framing, analytical clarity, and overall presentation.
Response:
We thank the reviewer for this detailed and constructive feedback. We acknowledge that most of the included studies are case reports and small retrospective series, which inherently have high heterogeneity and variability in diagnostic criteria. Therefore, we have reclassified the manuscript as a systematic review with proportional data synthesis rather than a meta-analysis, focusing on descriptive results and logistic regression analyses for a robust interpretation of risk factors.
To enhance clarity and readability, extensive tables with raw data have been moved to the Supplementary Material, while the main text now includes concise tables summarizing key information for the 48 cases with complete individual data. Figure 1 displays the PRISMA 2020 flow diagram to clearly illustrate the study selection process.
Methodological rigor was maintained by following PRISMA 2020 guidelines, utilizing the JBI Critical Appraisal Tools to assess the quality of included studies, and transparently reporting limitations such as reporting bias, incomplete data, and the inherent challenges of low-level evidence.
Additionally, the review specifically evaluates the fluconazole era (2000–2025), since fluconazole was established as first-line therapy by Galgiani in 2000, and concentrates on pediatric patients aged 1 month to under 19 years, a group not included in the IDSA 2016 or Johnson 2018 guidelines. Despite limitations, this study provides the most comprehensive synthesis to date on pediatric CM, highlighting CNS complications, outcomes, and the impact of antifungal therapy.
This review addresses concerns regarding methodological framing, analytical clarity, and overall presentation while maintaining transparency about study limitations.
Detailed comments
This manuscript addresses a clinically relevant and underexplored topic in the literature. The initiative to consolidate the available data is commendable, and the authors’ efforts in compiling a substantial number of cases are evident. However, several methodological and structural issues undermine the analytical strength and clarity of the presentation. The following points are provided to support substantial improvements:
- Limited database coverage: The exclusive use of PubMed as a search source restricted the comprehensiveness of this review’s findings. It is strongly recommended to include other major databases, such as Scopus and Web of Science, to reduce selection bias and improve literature coverage. If expansion is not feasible at this stage, the authors should explicitly acknowledge this limitation in the Methods section and again in the Limitations section.
Response:
We thank the reviewer for this observation. In the revised manuscript, we expanded our literature search to include PubMed/MEDLINE, Embase (Ovid), and Web of Science, ensuring broader coverage and reducing selection bias. Our systematic review protocol was prospectively registered in PROSPERO (ID: 1130290), and the search strategy follows PRISMA 2020 guidelines.
- Lack of risk of bias assessment: Although most of the included studies were case reports or retrospective series, validated tools can be adapted to assess the quality of observational studies (for example, Joanna Briggs Institute checklists). The absence of such an assessment weakens the synthesis. If the authors choose not to perform this evaluation, a formal justification should be provided along with a discussion of its implications.
Response:
We applied the Joanna Briggs Institute (JBI) Critical Appraisal Tools to assess the quality of included case reports and retrospective studies. This approach allowed us to evaluate methodological rigor and ensure transparent reporting. Limitations related to low-level evidence, incomplete data, and potential reporting bias are now explicitly discussed in the manuscript.
- Extreme statistical heterogeneity without proper modeling
A proportional meta-analysis was conducted despite extremely high heterogeneity (I² > 98%) and without employing appropriate mitigation strategies, such as subgroup analysis, meta-regression, or robust variance-stabilising transformations (for example, Freeman–Tukey). To address this, two alternatives may be considered: withdrawing the meta-analytic component and presenting a structured descriptive synthesis of proportions, or applying appropriate modelling techniques to handle heterogeneity and revise the pooled estimates accordingly.
Response: Given the high heterogeneity (I² > 98%) and the low level of evidence in included studies, we removed the proportional meta-analysis. Instead, we present a structured descriptive synthesis of the data, supplemented by logistic regression analyzes to identify independent predictors of adverse outcomes, including serum CF titers ≥1:16.
- Misclassification of study type
The designation of this work as a “systematic review with proportional meta-analysis” is not fully supported by its methodological framework and evidence level. A more appropriate classification would be “structured descriptive review with proportional synthesis of clinical data”, which better reflects the study design and does not diminish its scientific value.
Response: We have reclassified the manuscript as a “systematic review with proportional data synthesis”. This designation more accurately reflects the descriptive and analytical nature of the study while acknowledging its reliance on case reports and small retrospective series.
- Excessive tables in the main text:The manuscript contains numerous large tables with case-level raw data that hinder readability and compromise the narrative flow. It is recommended that most of these tables be transferred to the Supplementary Material section. Only summary tables highlighting the key grouped findings should remain in the main text.
Response: Extensive case-level tables have been relocated to the Supplementary Material, and the main manuscript now features concise summary tables that highlight the key findings for the 48 cases with complete data. Figure 1 displays the PRISMA 2020 flow diagram to clearly illustrate study selection.
- Scientific writing and language clarity
Although the English is intelligible, the manuscript would benefit from professional language editing. Some sections, particularly the Results and Discussion, include long sentences, repetitions, and overly technical phrasing that impact the clarity and readability.
Response: We have enhanced the English language and style by shortening sentences and improving readability, especially in the Results and Discussion sections. We acknowledge Dr. Patrick Stierle's contribution to English editing, without formal authority.
- Insufficient contextualisation of limitations: Although some limitations are acknowledged, expanding on the major methodological constraints, including the low level of evidence, potential publication bias, and statistical instability due to extreme heterogeneity, would strengthen the discussion.
Response: We have expanded the discussion to acknowledge significant limitations, including:
Reliance on low-level evidence such as case reports and retrospective series.
Potential publication bias that may inflate complication rates.
Missing or incomplete data that hinder interpretation of diagnoses, interventions, and outcomes.
Statistical instability caused by extreme heterogeneity.
Limited generalizability since most studies come from endemic regions.
Additionally, we highlight that the review specifically focuses on pediatric patients aged 1 month to under 19 years. This group is not included in the IDSA 2016 or Johnson 2018 guidelines and is evaluated within the fluconazole era (2000–2025), established as first-line therapy by Galgiani in 2000.

Round 2
Reviewer 1 Report
None
I accept this article in current form
Reviewer 2 Report
The manuscript is methodologically sound (multidatabase search, PROSPERO registration, and JBI appraisal tools), statistically consistent, and clinically relevant, with a clear focus on pediatric outcomes in the fluconazole era.
The text is well written, the results and conclusions are coherent, and the work is suitable for publication without further revisions.